# Considerations in evaluating equipment-free blood culture bottles: A short protocol for use in low-resource settings

Sien Ombelet[1,2], Alessandra Natale[3], Jean-Baptiste Ronat[3,4,5], Olivier Vandenberg[6,7,8], Jan Jacobs[1,2], Liselotte Hardy[1]*

**1** Department of Clinical Sciences, Institute of Tropical Medicine, Antwerp, Belgium, **2** Immunology & Microbiology Department, KU Leuven, Leuven, Belgium, **3** Médecins Sans Frontières, Paris, France, **4** Team ReSIST, INSERM U1184, School of Medicine University Paris-Saclay, Paris, France, **5** Bacteriology-Hygiene Unit, Assistance Publique – Hôpitaux de Paris, Bicêtre Hospital, Le Kremlin-Bicêtre, France, **6** Center for Environmental Health and Occupational Health, School of Public Health, Université Libre de Bruxelles (ULB), Brussels, Belgium, **7** Innovation and Business Development Unit, Laboratoire Hospitalier Universitaire de Bruxelles - Universitair Laboratorium Brussel (LHUB-ULB), ULB, Brussels, Belgium, **8** Division of Infection and Immunity, Faculty of Medical Sciences, University College London, London, United Kingdom

* lhardy@itg.be

**Data Availability Statement:** Relevant data are within the following repository: https://doi.org/10.6084/m9.figshare.14994057.v1.

## Abstract

Use of equipment-free, "manual" blood cultures is still widespread in low-resource settings, as requirements for implementation of automated systems are often not met. Quality of manual blood culture bottles currently on the market, however, is usually unknown. An acceptable quality in terms of yield and speed of growth can be ensured by evaluating the bottles using simulated blood cultures. In these experiments, bottles from different systems are inoculated in parallel with blood and a known quantity of bacteria. Based on literature review and personal experiences, we propose a short and practical protocol for an efficient evaluation of manual blood culture bottles, aimed at research or reference laboratories in low-resource settings. Recommendations include: (1) practical equivalence of horse blood and human blood; (2) a diverse selection of 10 to 20 micro-organisms to be tested (both slow- and fast-growing reference organisms); (3) evaluation of both adult and pediatric bottle formulations and blood volumes; (4) a minimum sample size of 120 bottles per bottle type; (5) a formal assessment of usability. Different testing scenarios for increasing levels of reliability are provided, along with practical tools such as worksheets and surveys that can be used by laboratories wishing to evaluate manual blood culture bottles.

## Introduction

Diagnosis of bloodstream infection (BSI) relies mainly on blood culture, *i.e.* the culture of large volumes of blood in blood culture bottles (BCB). In low-resource settings (LRS), mainly so-called "manual" blood culture systems are used, as opposed to the automated blood culture systems standardly used in high-resource settings. Manual BCBs are incubated in a

**Funding:** This work was partly funded by Médecins Sans Frontières (Operational Centre Paris). The funder participated in study design and preparation of the manuscript.

**Competing interests:** The authors have declared that no competing interests exist.

conventional incubator, detection of growth is done based on visual signs of growth [1]. According to a market forecasting study, manual blood culture systems will make up roughly two-thirds of the global blood culture market by 2025 [2]. It is therefore surprising that almost no research has been done on manual blood culture systems since the 1990's, when automated systems were introduced. For a district hospital laboratory in LRS, it is difficult to find scientific guidance when looking for the optimal BCB brand. The use of home-made BCB is widespread in LRS but should be discouraged due to lack of quality assurance of production methods and the omission of necessary additives [1, 3]. Import of commercial BCB into LRS countries is often expensive and logistically difficult. To address these problems, some LRS reference laboratories have started local, centralized production of blood culture media–an example is the Centralized Media Making Laboratory in Cambodia (https://dmdp.org/media-making-lab/). To ensure acceptable performance of such BCBs, and of newly marketed commercial manual BCBs, it is necessary to perform a validation study, in which these BCBs are preferably compared to the "gold standard" automated blood culture system.

BCB performance is evaluated using phase III clinical comparative studies, in which new BCBs are compared with the gold standard in patients requiring blood culture. However, randomization between blood culture systems requires hypothetic equivalence of both systems, and automated systems have long shown to be superior to manual blood culture systems in terms of yield and time-to-detection [1]. Another option is to inoculate half of the patient's blood in each system, but this may also deprive the patient of an essential diagnosis, as larger volumes of blood increase the yield of blood cultures [4]. Sampling more blood from patients to mitigate this risk also raises ethical issues, especially in children or patients with low hemoglobin levels [5]. Another argument against clinical comparison studies is the long duration of this study type and the low yield of blood cultures; 85–95% of sampled blood cultures will not show growth, therefore clinical studies require very high sample sizes (typically 8.000 to 9.000) to detect a sufficient number of rare pathogens [6]. Laboratory studies, in which BCBs are spiked with blood and microorganisms causing BSI, can be a valuable alternative. These i*n vitro* studies have no implications on patient management and are more efficient; the BCBs can be spiked with a variety of organisms, pre-defined by the researchers, guaranteeing sufficient numbers and inclusion of all desired pathogens.

In this manuscript, we list some "best practices" for simulated blood cultures, based on literature review and lessons learned from studies carried out at our research group from 2017 to 2020. We propose a practical protocol for a short and efficient validation of manual BCBs using simulated blood cultures and an automated "reference" system. This validation assures minimal performance of the BCB under evaluation, for settings where large studies are not feasible.

## BCB performance evaluation study: Set-up, considerations and recommendations

### Preparing a BCB performance evaluation study

**Objectives of the BCB evaluation study.**    The main objective of the BCB evaluation study is to compare yield of a new manual BCB with an automated reference system. A secondary objective is to compare speed of growth between systems. Other secondary objectives are assessment of inter-lot variability and usability of BCBs. Some definitions used in this protocol are given in Table 1.

**Acceptability criteria need to be determined before starting the experiments.**    These criteria will depend on practical considerations, such as price and availability of other BCB and expected patient population. We developed our own set of acceptability criteria for a BCB

**Table 1. Definitions of terms used in the protocol proposal.**

| Term used | Definition |
|---|---|
| **Run** | BCB inoculated in triplicate with the same strain simultaneously and with the same bacterial suspension. The same strain can be tested in different runs to assess inter-lot variability. |
| **Growth** | BCB showing signs of growth, either visual (e.g. turbidity, hemolysis or gas production) or as flagged by the automate, with confirmation of bacterial growth by subculture on agar (colonies of the inoculated organism) or by microscopy (bacteria visible on the Gram stain). Bottles showing signs of growth that cannot be confirmed by microscopy or subculture (48h incubation in the case of fastidious organisms and yeast), should be regarded as false-positive. |
| **Yield** | Percentage of inoculated bottles that show confirmed growth, as defined above |
| **Speed of growth** | Defined as time-to-detection in hours for automated systems, and for cumulative yield on day 1 and day 2 of incubation for manual bottles. |
| **Day of incubation** | A BCB is inoculated on day 0 of incubation. Day 1 of incubation is the day after first overnight incubation. Day 2 of incubation is after two nights of incubation. |
| **Blind subculture** | A subculture of the blood/broth mixture on an agar plate regardless of the presence of visual signs of growth |

Abbreviation: BCB = blood culture bottle.

evaluation for the Mini-lab project [7] (Table 2). In addition, Dailey *et al.* formulated an extensive target product profile for a blood culture system in LRS [3]. The most important acceptability criteria are yield, speed of growth and ease of use of the BCB under evaluation.

## Three testing scenario's: From minimal to optimal

We propose three "scenarios", depending on how extensive the validation needs to be (Table 3). For a minimal scenario, 10 different bacterial species are tested (Table 4) with 1 strain per species. Both BCB types (BCB under evaluation and reference system) are inoculated in triplicate with the same blood-bacteria suspension (= one run). This is done once for adult formulations, once for pediatric formulations, with corresponding volumes of blood and representative bacterial concentrations. These runs are then repeated for a second lot of BCBs, to assess inter-lot variability. For the optimal scenario, 20 species are tested in total for three lots, for both adult and pediatric formulations. In the intermediate scenario, 20 species are tested for two lots. Table 3 gives an overview of number of BCBs and volume of blood needed for each of these scenarios. For pathogen species showing suboptimal growth in the BCB under evaluation (yield of less than 75%), we recommend testing additional strains of the same species; the extra number of BCBs and volume of blood needed in this case are also given in Table 3.

We performed basic sample size calculations to estimate the level of differences that can be detected with these testing scenarios (Table 3). For these sample size calculations, we assumed two-sided tests, confidence levels of 95%, power levels of 80%, and a yield of the reference

**Table 2. Criteria for blood culture bottle performance, as defined by the mini-lab project.**

| Weight of importance | Criteria |
|---|---|
| 50% | The yield in the bottle tested should **not be less than 90%** of the yield in the reference method (automated blood culture system) |
| 40% | Cumulative % of grown bottles should be at least **75% at day 1 of incubation and 90% at day 2 of incubation** |
| 10% | There should be **no systematic failure** in any of the non-fastidious species tested (systematic failure = growth in less than 75% of BCB, compared to reference method) |

**Table 3. Logistical and statistical implications of three scenarios of blood culture bottle (BCB) validation.**

| | Optimal scenario | Intermediate scenario | Minimal scenario |
|---|---|---|---|
| Number of strains tested | 20 | 20 | 10 |
| Number of lots tested | 3 | 2 | 2 |
| Total number of bottles needed per bottle type | 360 | 240 | 120 |
| Total number of bottles needed (all bottle types combined) | 720 | 480 | 240 |
| Total volume of blood needed | 4320 ml | 2880 ml | 1440 ml |
| Number per bottle type adult | 180 | 120 | 60 |
| Number of bottle type pediatric | 180 | 120 | 60 |
| Detectable difference in yield * | 5% | 7% | 10% |
| Detectable relative yield * | 95% | 93% | 90% |
| 95% confidence interval ** | 86–93% | 85–93% | 83–95% |
| Number of extra bottles needed per extra strain tested per bottle type | 18 | 12 | 12 |
| Volume of extra blood needed per extra strain tested | 216 | 144 | 144 |

Calculation of detectable differences in yield is based on the normal approximation of the binomial distribution (https://www.stat.ubc.ca/~rollin/stats/ssize/b2.html); confidence intervals are calculated based on the binomial distribution (http://vassarstats.net/prop1.html).

* Compared to reference system, assuming 80% power, 95% confidence and 97% yield of the reference system

** Assuming an observed yield of BCB under evaluation of 90%; confidence interval becomes narrower when observed yield of BCB under evaluation is higher.

Confidence intervals for proportions are not symmetrical due to binomial distribution; the uncertainty for these proportions is larger on the lower side of the interval than on the higher side.

**Table 4. Proposal for species to be tested in a blood culture bottle performance validation study in low-resource settings.**

| Pathogen group | Species | Speed of growth |
|---|---|---|
| Enterobacterales | *Escherichia coli* | Fast |
| | *Salmonella* Typhimurium | Fast |
| | *Klebsiella pneumoniae* | Fast |
| | *Salmonella* Typhi* | Fast |
| | *Enterobacter cloacae** | Fast |
| Non-fermenters | *Pseudomonas aeruginosa* | Intermediate |
| | *Acinetobacter baumannii* | Slow |
| | *Burkholderia cepacia** | Slow |
| Staphylococcus | *Staphylococcus aureus* | Intermediate |
| | *Staphylococcus epidermidis** | Slow |
| Streptococcus | *Streptococcus pneumoniae* | Fast |
| | *Streptococcus pyogenes** | Fast |
| | *Streptococcus anginosus** | Intermediate |
| | *Streptococcus suis** | Intermediate |
| Fastidious organisms | *Haemophilus influenzae* | Slow |
| | *Neisseria meningitidis* | Slow |
| Enterococcus | *Enterococcus faecalis** | Intermediate |
| Yeast | *Candida albicans* | Slow |
| | *Cryptococcus neoformans** | Slow |
| | *Candida tropicalis** | Slow |

*only to be included in the "optimal scenario"

system of 97% (based on unpublished data from our research group and yield of automated systems published in literature) [8–10]). An online calculator was used for the sample size calculation (https://www.stat.ubc.ca/~rollin/stats/ssize/b2.html) and the VassarStats website for calculation of the confidence interval (http://vassarstats.net/prop1.html). To detect a difference in yield of 10% between the BCB under evaluation and the reference system, corresponding to a relative yield of 90% compared to the reference system, a sample size in each group of at least 115 is needed. This would correspond to a "minimal" testing scenario (Table 3). Such a "minimal" scenario avoids unnecessary extensive validation of a BCB which has a clearly suboptimal performance (< 90% of the reference system). Some of the assumptions of these calculations will influence detectable differences. The higher the yield of the reference system, the higher the power to detect small differences. Stratifying results according to strain, adult versus pediatric formulation or lot number, on the other hand, will decrease power. Appropriate data analysis (taking into account the dependence of the data) may further increase power [11].

A negative control for each BCB type should be added to the test panel per run, to ensure sterility of the blood used and of the procedures. The negative control is a BCB filled with non-spiked blood from the same blood bag or lot of horse blood as used for the experiments.

## Performance evaluation

**The use of horse blood has logistical and ethical advantages compared to human blood.** Most simulated BCB studies are done with human banked blood [10, 12, 13–21], because differences in plasma components and red blood cell enzymes between mammal species may affect bacterial growth. In addition, variations in visible signs of growth are possible due to inter-species differences in aggregation characteristics and fragility of red blood cells [22–24]. Despite these advantages, the large volumes of blood needed (> 4 liter, see Table 3) imply ethical and logistic difficulties. The availability of left-over blood in blood banks is limited and variable over time, jeopardizing efficient experiment planning. Left-over blood from multiple volunteers is pooled, compromising standardization.

An alternative to human blood is defibrinated horse blood [8, 25–31]. Altun *et al.* claimed that using horse blood does not significantly alter the performance of blood culture systems [8]. However, no evaluations of equivalence between human blood and horse blood used for BCB evaluation have been published. Advantages of horse blood over human blood are availability, logistic ease and standardization, at least in those settings where commercial horse blood is available. To demonstrate equivalence of horse blood for use in BCB evaluation studies, we performed a head-to-head comparison of horse and human blood (S1 File). We demonstrated that horse blood can be used for BCB validation studies. Sheep blood is generally more available in LRS [32]. Few BCB validation studies using sheep blood have been published [33]. Before systematically using sheep blood, a validation study is needed. To ensure equivalent visual and growth characteristics, packed cell volume of both commercial horse and sheep blood should be similar to human blood (35–45%).

**A diverse selection of clinically relevant micro-organisms should be tested.** To evaluate whether all types of pathogens grow well in the BCB under evaluation, a broad selection of invasive pathogens should be used. Evidently, key BSI pathogens, depending on the setting, should be selected. We propose a list of essential organisms tailored to LRS in Table 4. This list includes all important pathogen "groups" and both fast- and slow-growing microorganisms [17, 34–38]. Fast-growing organisms have a median time-to-positivity in automated systems between approximately 9–15 hours, whereas slow-growing organisms have time-to-positivity > 18 hours. We only recommend testing of aerobic bacteria (and BCB), as anaerobic bacteremia is less relevant in LRS [1].

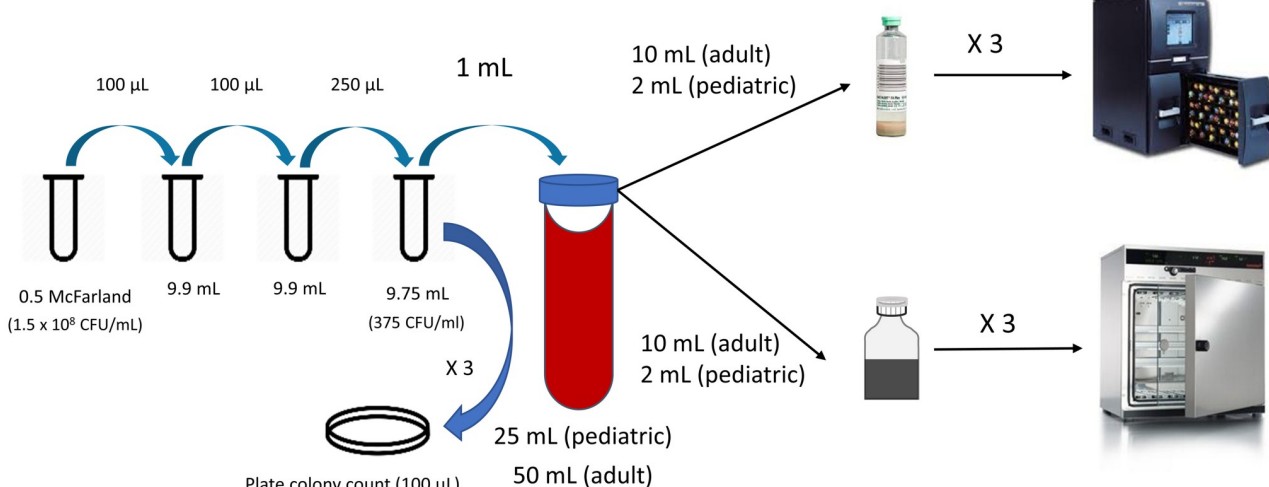

**Fig 1. Recommended methods for spiking blood culture bottles.** Calibrated pipettes must be used. Strains to be tested undergo two passages on blood or chocolate agar before the suspension is prepared. To confirm the concentration of the final bacterial suspension, 100 μl spread out on a blood agar plate for colony counts the next day (in triplicate). CFU = colony forming units.

We advise using certified bacterial strains (*e.g.*, American Type Culture Collection (ATCC)) instead of clinical strains, as these are representative members of their species. Reproducibility and standardization of the study results is therefore expected. However, basic growth characteristics of the strains used should be assured: some ATCC strains are known to grow poorly in certain culture media (*e.g.*, *Escherichia coli* ATCC 8739, https://www.lgcstandards-atcc.org/products/all/8739.aspx?geo_country=be#culturemethod); the use of such strains is best avoided.

**Dilution series can be used to achieve low bacterial concentrations.** The concentration of micro-organisms in the blood of adult BSI patients is in the order of <1–10 colony forming units (CFU) per mL [39–42]. In clinical blood cultures, a BCB filled with 10 mL of blood contains on average 1–100 CFUs. This range should be respected in *in vitro* BCB evaluations. Indeed, most studies using simulated blood cultures add between 5 and 125 CFU per BCB [8, 12–16, 19–21, 29]. A low CFU concentration can be achieved by making serial dilutions from a 0.5 McFarland suspension containing approximately $1.5 \times 10^8$ CFU/mL [43], at least for Gram-negative bacteria such as *E. coli*. This is not true for all microorganisms: *e.g.* for *Candida* species, 0.5 McFarland only contains $1–5 \times 10^6$ CFU/mL [44]. Dilution series should take this into account [17]. Methods for dilution series are depicted in Fig 1.

Plate colony counts to check bacterial concentrations are common practice, but have a low precision due to sampling variation, especially at low concentrations [45–49]. The mean colony count follows the Poisson distribution, implying that the lower the concentration, the higher the standard deviation relative to the mean, and the less reliable the plate count will turn out to be. The same sampling variation applies to adding colonies to BCBs. Reliably counting (or adding) CFU numbers lower than 25 is challenging; this is considered the "limit of quantification" [45]. To mitigate the effect of the large total error on the estimated bacteria concentration of the suspension, it is generally advised to take the mean of three plate counts of the same suspension [45]. Triplicate plate counts are also advised as a quality check of the dilution series; in case of persistently lower or higher counts than expected, or in case of large variation between the three separate plate counts per dilution, the dilution procedure should

be adapted. Plate counts are also useful in the phase of data analysis, as higher initial bacterial inocula could lead to faster growth and must be adjusted for.

**Both adult and pediatric formulations should be tested.** Bacterial concentrations in children with bacteremia are believed to be higher than in adults, but blood volumes added to the BCB are lower [50, 51]. Testing of both adult and pediatric blood volumes and bacterial concentrations is advised, as different blood-broth ratios can affect performance of the BCBs [52]. Moreover, many manufacturers have different BCBs for adult and pediatric blood cultures. Blood volume added to pediatric BCBs in simulated blood culture studies is usually 4–5 mL [16, 19–21]. This is the upper limit of recommended blood volumes in children, but is not entirely representative for pediatric populations, as sampled blood volumes typically vary between 0.5 mL (for neonates) and 4 mL (for older children) [51]. We therefore recommend using a volume of 2 mL. For adult formulations, a volume between 8 and 10 mL per BCB is often used [8, 12, 13, 16, 19–21, 29] and is indeed recommended in clinical practice [4].

**Growth in manual BCBs should be visually assessed twice daily.** Before use, each BCB should be checked visually for evident signs of contamination, such as broth turbidity or colonies on agar slant (for biphasic BCBs). After inoculation, the reference BCBs are incubated until the automate detects growth. Manual BCBs are inspected twice daily for visual signs of growth, such as turbidity, hemolysis and gas production [1] (Fig 2). To standardize this visual inspection, it is advisable to use a diffuse light source, such as a lightbox. The cycle of incubation/inspection continues until the broth (or agar slant) shows visual signs of growth, or up to 7 nights of incubation in case of no visible growth. For all manual BCBs, a blind subculture is

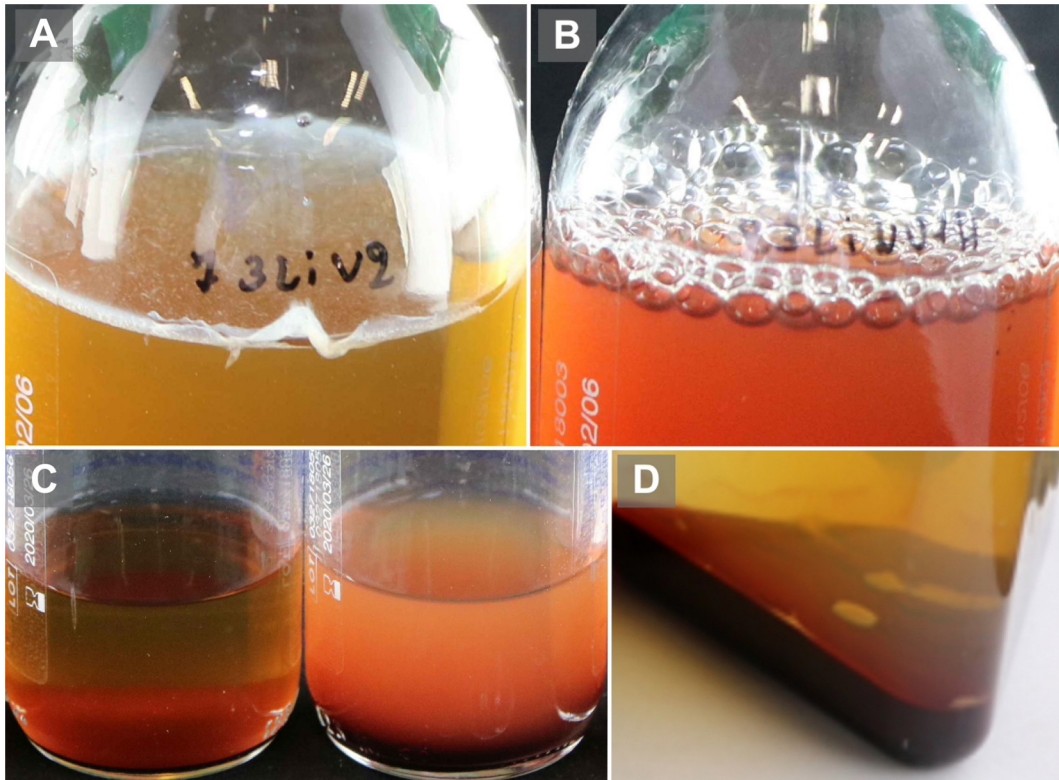

**Fig 2. Visual signs of growth in manual blood culture bottles.** (A): pellicle formation on surface; (B) gas production; (C) turbidity (left bottle: no growth; right bottle: turbidity; (D) puff balls. (first published in Frontiers in Medicine (1), reproduced with permission).

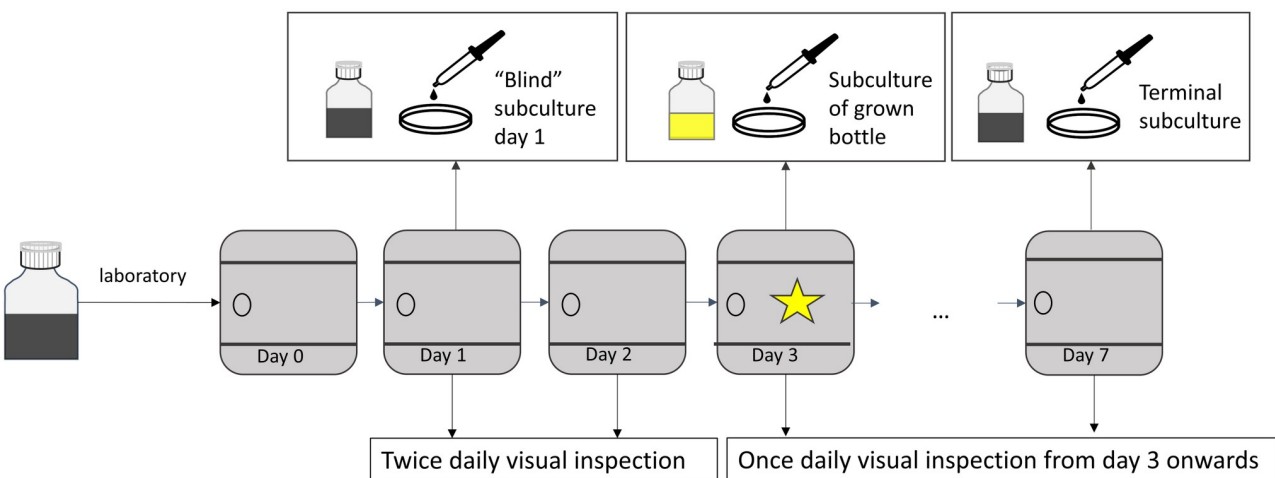

**Fig 3. Laboratory processing of spiked manual blood culture bottles.** The bottles are incubated on day 0. Blind subculture is performed on day 1, regardless of signs of growth. If blind subculture is negative, another subculture is done when growth is detected visually (in this example, day 3). If none of the subcultures shows growth, a terminal subculture is done on day 7.

performed on blood agar on day 1 (20–24 hours after start of incubation). A second subculture is done for manual BCBs with negative day 1 subculture, either upon visual growth ("positivity" subculture) or after 7 nights without visual growth ("terminal" subculture) to exclude growth (Fig 3). Autolysis of bacteria, leading to visual signs of growth but lack of colony growth on subculture, can be detected by performing Gram stain on such BCBs. A daily schedule of activities is presented in Table 5. Worksheet examples to be used during the study can be found in S2 File.

**Table 5. Daily schedule of activities during a week of BCB evaluation.**

| Day of incubation | Comment | Inspection/action |
|---|---|---|
| **0** | Day of spiking and incubation of the BCB | Preparation dilution series |
| | | Spiking of blood |
| | | Inoculation of BCB |
| | | Incubation in automated and static incubator (preferably before 12:00 h) |
| **1** | After 1 overnight incubation | *Morning*: inspect manual BCB for signs of growth |
| | | Perform blind subculture of manual bottles on blood agar |
| | | *Afternoon*: inspect manual BCB for signs of growth |
| **2** | After 2x overnight incubation | *Morning*: inspect manual BCB for signs of growth |
| | | *Afternoon*: inspect manual BCB for signs of growth |
| **3** | After 3x overnight incubation | Inspect for signs of growth |
| **4** | After 4x overnight incubation | Inspect for signs of growth |
| **5** | After 5x overnight incubation | Inspect for signs of growth |
| **6** | After 6x overnight incubation | Inspect for signs of growth |
| **7** | After 7x overnight incubation | Inspect for signs of growth |
| | | If no signs of growth and no colonies on blind subculture done at day 1: perform terminal subculture on blood agar |

**Data analysis should take "clustering" on bacterial strain into account.** Statistical methods to compare the yield and speed of growth of the manual BCB under evaluation with the reference system must take dependence of the data into account: the same bacterial suspension is used to inoculate three BCBs of both BCB types. Yield and speed of growth will be more similar among BCBs inoculated with the same suspension. The data is thus essentially "clustered" on the strain tested. Data analysis should take this into account, as this will affect confidence intervals and p-values for hypothesis tests; the direction of this change is dependent on the type of clustered data [11]. Several analysis approaches are possible; we refer to other papers for detailed information and examples [11, 53]. A practical and robust approach, in our opinion, would be to use a random effects logistic regression, as regression models have the ability to incorporate other variables as well [11].

## Ease of use assessment of BCB

Ease of use assessment is often overlooked in the evaluation of *in vitro* diagnostics (IVD), despite its importance in succesfull implementation of diagnostics in remote settings and WHO recommendations for IVD development [54, 55]. Apart from good technical performance, an ideal BCB allows easy sampling, by assuring a strong vacuum, a septum that is not too rigorous and compatibility with adapters for use with butterfly needles. Visual evaluation of growth for manual BCBs is facilitated by transparency of the bottle material, transparent labels and visual aids such as growth indicators. BCBs should allow easy labelling, with sufficient space to provide all necessary information. Lastly, BCB material should be in line with biosafety requirements. This pertains for instance to the possibility to safely autoclave the material before transport for incineration and subsequent disposal, as some types of plastic cannot withstand high temperatures. For more information on ease of use, see S3 File.

### Technical assessment of ease of use: A strong vacuum in the BCB facilitates sampling

The BCB vacuum allows for passive blood aspiration into the BCB. This is necessary when using butterfly needles, but will also facilitate sampling with syringe and needle as it obviates the need to put pressure on the syringe, an action that increases risk of needlestick injury and spills. Commonly used BCBs such as BacT/ALERT have a vacuum exceeding 10 mL, which enables easy sampling but entails a risk of overfilling [56]. To exactly measure the vacuum, specialized devices are needed, such as hydrostatic gauges or mercury column manometers [57]. However, this level of detail is not needed to assess ease of blood collection.

We propose a practical vacuum estimation based on passive aspiration of water, using 10 BCBs from at least 2 different lots. For the needle and syringe sampling method, vacuum is tested by connecting a 21-gauge needle to a 10 mL syringe filled with distilled water. When the needle perforates the BCB septum, the water is aspirated into the BCB (without applying pressure!). The volume of water still present in the syringe at the end of the procedure can be used to calculate the volume of water aspirated. For the butterfly needle sampling method, the hub and tube of a 21-gauge butterfly needle are put into a 100 mL measuring glass filled with distilled water. The septum of the BCB is pierced with the butterfly needle while the other end is kept stable in the measuring glass. After filling of the BCB by the vacuum, the difference between start and end volume of water in the measuring glass is measured and calculated. Using these test data, inter-lot and intra-lot coefficients of variation (CV) of BCB vacuum can be calculated, giving an indication of uniformity of production procedures.

#### Assessing ease of use by the end-user: Survey (S3 File)

In S3 File, we propose a survey for a qualitative assessment of ease of use. More information on essential aspects of ease of use can also be found in this Supporting information.

### Limitation of the study design

General limitations of simulated blood cultures apply here; this *in vitro* approach does not constitute a "real-life" situation. Many important variables that influence growth of bacteria, such as concentration of bacteria and the presence of antibiotics or other microbiocidal factors in the plasma, may substantially differ from clinical conditions. For manual BCB, which are visually inspected daily, there is also an important observer bias related to simulated blood cultures; the laboratory technicians know that BCBs have been inoculated with microorganisms and this can influence their assessment of signs of growth. Blinded assessment is possible but implies inoculating a large number of BCBs with non-spiked blood as well; this would dramatically increase workload and volume of blood needed. Because of these limitations, results from simulated blood culture studies cannot be directly generalized to the field.

Most studies first add a bacterial suspension with a pre-defined number of CFU to the BCB and then add blood. We opted for another approach, adding the micro-organisms to the blood first and inoculating the blood into the BCB after homogenization of the blood-microorganism mixture on a roller-mixer. Introducing the bacterial suspension directly into the BCB requires inoculation of very small volumes (100 μL) with syringe and needle. This is difficult to do reliably, with the risk of considerably increasing variation in CFU between BCBs, especially as relatively low concentrations are used. Moreover, spiking the blood first creates a situation more similar to real-life conditions, as differences in CFU added between BCBs will depend on the sampling variation of the eventual concentration of bacteria in the spiked blood, and less on technical imprecisions during inoculation. However, this approach adds an additional dilution step with its associated additional variation.

Vacuum in BCBs is not static and typically decreases over time. When comparing the vacuum in BCBs from different lot numbers, it is therefore important to take the date of production into account and to compare only BCBs of approximately the same age (inferred from the expiration and production date). Moreover, many BCBs have a vacuum that aspirates more than 10 mL of water. The proposed method is only appropriate to assess whether vacuum is sufficient to easily aspirate blood, not to give reliable means for vacuums exceeding passive aspiration of 10 mL. Lastly, viscosity of water is lower than that of blood; therefore, the volume of water passively aspirated may be higher than the volume of blood passively aspirated.

### Conclusion and future research possibilities

This report aims to give practical guidance to laboratories or research facilities that wish to evaluate the performance of (manual) BCB. We hope to encourage researchers and reference laboratories from LRS to validate BCB in their settings, to ensure acceptable quality of the BCBs and to add research on new manual BCB to the existing literature. There are no recent publications on performance of manual BCB, despite favorable market forecasts for these products [2]. To encourage clinical bacteriology at district laboratory level, quality assurance for consumables is of utmost importance [58]. Evaluation of ease of use of BCB and compatibility with common waste disposal practices such as autoclavation is seldom done, if at all. Nonetheless, these characteristics play an important role in successful implementation of blood cultures in LRS.

For many of the recommendations given above, empirical evidence from published studies was scarce. For example, we found no studies that compared horse blood with human blood

for validation studies. Another knowledge gap is the effectiveness of dilution series to assure appropriate bacterial concentrations; we found plate colony counts resulting from dilution series to be widely divergent across bacterial species. Studies quantifying differences in bacterial concentrations between species would be welcomed to obtain more precise estimates of number of CFU added to the BCB, as would improvements in the methods for quantification of bacterial concentrations. More research on best practices for performing standardized and efficient validation studies is needed.

## Supporting information

**S1 File. Report on head-to-head comparison of horse blood with human blood for blood culture bottle validation studies.**
(DOCX)

**S2 File. Examples of worksheets to be used in blood culture bottle validation study.**
(DOCX)

**S3 File. Ease of use assessment.**
(DOCX)

## Acknowledgments

We would like to thank Thierry Naas, Kaat Eggermont, Tine Vermoesen and Ellen Corsmit for their valued input and support in optimizing our blood culture validation protocol.

## Author Contributions

**Conceptualization:** Sien Ombelet, Alessandra Natale, Jean-Baptiste Ronat, Olivier Vandenberg, Jan Jacobs, Liselotte Hardy.

**Formal analysis:** Sien Ombelet.

**Funding acquisition:** Alessandra Natale, Jean-Baptiste Ronat.

**Methodology:** Sien Ombelet, Alessandra Natale, Jan Jacobs, Liselotte Hardy.

**Project administration:** Sien Ombelet.

**Supervision:** Olivier Vandenberg, Jan Jacobs, Liselotte Hardy.

**Visualization:** Sien Ombelet, Jan Jacobs, Liselotte Hardy.

**Writing – original draft:** Sien Ombelet.

**Writing – review & editing:** Alessandra Natale, Jean-Baptiste Ronat, Olivier Vandenberg, Jan Jacobs, Liselotte Hardy.

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
