## [Decision Letter · Decision Letter 0]

18 Mar 2022

PONE-D-21-35594Considerations in evaluating equipment-free blood culture bottles: a short protocol for use in low-resource settingsPLOS ONE

Dear Dr. Hardy,

Thank you for submitting your manuscript to PLOS ONE. After careful consideration, we feel that it has merit but does not fully meet PLOS ONE’s publication criteria as it currently stands. Therefore, we invite you to submit a revised version of the manuscript that addresses the points raised during the review process.

We look forward to receiving your revised manuscript.

Kind regards,

Mehmet Demirci, PhD

Academic Editor

PLOS ONE

Journal Requirements:

3. Please ensure that you include a title page within your main document. You should list all authors and all affiliations as per our author instructions and clearly indicate the corresponding author

Additional Editor Comments (if provided):

Comments from reviewer 1 (also found below):

The present manuscrpit proposes a short and practical protocol for an efficient evaluation of manual blood cultures bottles. It is well written and the protocol is very clear and detailed which makes it easy to use. Figures 1 and 3, which are well constructed, three "scenarios" as well as worksheets proposed in Supplementary Material 2 were much appreciated.

I have, however, some comments to the manuscript:

1- Table 1: "visual signs of growth" (turbidity, hemolysis and gas production line 191-192) should be detailled in Table 1 for a better understanding.

2- Table 1: Why make a methylene blue stain ? What is the added value compared to the Gram stain ?

3- Could Figures 1 and 2 be presented with better file resolution? Examples of visual signs of growth shown in figure 2 are not easy to see with this file resolution.

4- Figure 1 legend: Please explain why "strains to be tested undergo TWO passages on [...] agar before the suspension is prepared".

5- " In clinical blood cultures, a BCB [...] contains on average 1-100 CFUs. This range should be respected". Why does the method illustrated in Figure 1 recommand the use of 375 CFUs ? Is it comparable to a "real-life" situation ?

6- Line 189: "Before use, each BCB should be checked visually for evident signs of contamination". Is the visual control sufficient? Shouldn't subculture be performed before inoculation of the BCB to ensure their sterility?

Reviewers' comments:

Reviewer's Responses to Questions

**Comments to the Author**

1. Does the manuscript report a protocol which is of utility to the research community and adds value to the published literature?

Reviewer #1: Yes

Reviewer #2: Yes

2. Has the protocol been described in sufficient detail?

Descriptions of methods and reagents contained in the step-by-step protocol should be reported in sufficient detail for another researcher to reproduce all experiments and analyses. The protocol should describe the appropriate controls, sample sizes and replication needed to ensure that the data are robust and reproducible.

Reviewer #1: Yes

Reviewer #2: Yes

3. Does the protocol describe a validated method?

Reviewer #1: Yes

Reviewer #2: Yes

4. If the manuscript contains new data, have the authors made this data fully available?

Reviewer #1: Yes

Reviewer #2: Yes

**5. Is the article presented in an intelligible fashion and written in standard English?**

Reviewer #1: Yes

Reviewer #2: Yes

6. Review Comments to the Author

Reviewer #1: The present manuscrpit proposes a short and practical protocol for an efficient evaluation of manual blood cultures bottles. It is well written and the protocol is very clear and detailed which makes it easy to use. Figures 1 and 3, which are well constructed, three "scenarios" as well as worksheets proposed in Supplementary Material 2 were much appreciated.

I have, however, some comments to the manuscript:

1- Table 1: "visual signs of growth" (turbidity, hemolysis and gas production line 191-192) should be detailled in Table 1 for a better understanding.

2- Table 1: Why make a methylene blue stain ? What is the added value compared to the Gram stain ?

3- Could Figures 1 and 2 be presented with better file resolution? Examples of visual signs of growth shown in figure 2 are not easy to see with this file resolution.

4- Figure 1 legend: Please explain why "strains to be tested undergo TWO passages on [...] agar before the suspension is prepared".

5- " In clinical blood cultures, a BCB [...] contains on average 1-100 CFUs. This range should be respected". Why does the method illustrated in Figure 1 recommand the use of 375 CFUs ? Is it comparable to a "real-life" situation ?

6- Line 189: "Before use, each BCB should be checked visually for evident signs of contamination". Is the visual control sufficient? Shouldn't subculture be performed before inoculation of the BCB to ensure their sterility?

Reviewer #2: This Lab protocol paper will be an useful guideline for blood culture. My only concern is about your previous publication (PMID: 31275940) which is a more comprehensive paper in this regard. I understand that the submitted paper focus is protocol but it has overlapping with your mentioned publication.

7. PLOS authors have the option to publish the peer review history of their article (what does this mean?). If published, this will include your full peer review and any attached files.

Reviewer #1: No

Reviewer #2: **Yes: **Farzam Vaziri

---

## [Author Response · Author response to Decision Letter 0]

29 Mar 2022

Dear editors,

First of all, we would like to thank the editor and all reviewers for their time and interest in this manuscript, and for the useful suggestions.

We will respond to the comments raised by the reviewers one by one. We have kept the original comments in black and our responses are given in blue.

Comments for first revision, requested 28-03-22 by the editor and reviewers 18-03-22

Reviewer #1: 

1- Table 1: "visual signs of growth" (turbidity, hemolysis and gas production line 191-192) should be detailed in Table 1 for a better understanding.

We have indeed added some lines on characteristics of visual signs of growth in Table 1 (line 81 in the revised manuscript). 

2- Table 1: Why make a methylene blue stain ? What is the added value compared to the Gram stain ?

We have removed the mention of methylene blue stain in Table 1; this was a practice we did in our laboratory for a specific study but it is indeed not recommended in general for blood culture bottle validation studies.

3- Could Figures 1 and 2 be presented with better file resolution? Examples of visual signs of growth shown in figure 2 are not easy to see with this file resolution.

We have attempted optimal file resolution in our re-submission.

4- Figure 1 legend: Please explain why "strains to be tested undergo TWO passages on [...] agar before the suspension is prepared".

This was done in line with our laboratory procedures for subculturing of quality control strains, to ensure optimal viability and purity of the strain. Sparse growth after subculture from frozen strains and possible contamination by other strains can be overcome by subculturing twice. This procedure was based on recommendations by the Clinical Microbiology Procedures Handbook (Leber et al, 2016) for using frozen cultures as quality control for antibiotic susceptibility testing. We could not find in literature or handbooks whether this step is needed for validation of blood cultures bottles. We can therefore not make firm recommendations but only explained how it was done in our laboratory. 

5- " In clinical blood cultures, a BCB [...] contains on average 1-100 CFUs. This range should be respected". Why does the method illustrated in Figure 1 recommend the use of 375 CFUs ? Is it comparable to a "real-life" situation ?

From the suspension of 375 CFU/mL, 1 ml (=375 CFU) is added to 25 mL (pediatric) or 50 mL (adult) volume of blood. Then this volume of blood is further subdivided over the bottles at a volume of 10 mL (adult) or 2 mL (pediatric) per bottle. Therefore, the eventual number of CFU per bottle is 375/50*10 for adults and 375/25*2 for children. This comes down to 75 CFU in total for an adult bottle (concentration 7.5 CFU/mL) and 30 CFU for a child (concentration 15 CFU/mL). Both of these quantities are in the range of 1-100 CFU per bottle. 

6- Line 189: "Before use, each BCB should be checked visually for evident signs of contamination". Is the visual control sufficient? Shouldn't subculture be performed before inoculation of the BCB to ensure their sterility?

Sterility control of all blood culture bottles would be time-consuming and labor-intensive and risks actually contaminating the bottle in the process (although that risk is admittedly low). When the bottles are incubated and show growth, subculture is done and the colony aspect and Gram stain are used to ensure the grown micro-organisms are consistent with the spiked pathogens. 

In a previous version of this manuscript, we had advised to perform a sterility control on a subset of each lot of bottles to be evaluated by incubating them for 7 days and then performing a subculture of all of these bottles; however, at an expected contamination rate of < 5% of all bottles, the number needed to reliably rule out contamination would be too high to be feasible. 

Reviewer #2: This Lab protocol paper will be an useful guideline for blood culture. My only concern is about your previous publication (PMID: 31275940) which is a more comprehensive paper in this regard. I understand that the submitted paper focus is protocol but it has overlapping with your mentioned publication.

The paper you mention is a general review on use of blood cultures, whereas this manuscript specifically looks at validation of (manual) blood culture bottles to ensure their performance. With regards to some aspects (explanation of bacterial concentration, workflow, visual signs of growth), we agree there is some overlap, however we believe the aim and content of both papers is sufficiently different to justify separate publications.

Kind regards,

Liselotte

---

## [Editor Report · Decision Letter 1]

11 Apr 2022

Considerations in evaluating equipment-free blood culture bottles: a short protocol for use in low-resource settings

PONE-D-21-35594R1

Dear Dr. Hardy,

We’re pleased to inform you that your manuscript has been judged scientifically suitable for publication and will be formally accepted for publication once it meets all outstanding technical requirements.

Kind regards,

Assoc. Prof. Mehmet Demirci, PhD

Academic Editor

PLOS ONE
---

## [Editor Report · Acceptance letter]

13 Apr 2022

PONE-D-21-35594R1 

Considerations in evaluating equipment-free blood culture bottles: a short protocol for use in low-resource settings 

Dear Dr. Hardy:

I'm pleased to inform you that your manuscript has been deemed suitable for publication in PLOS ONE. Congratulations! Your manuscript is now with our production department. 

Kind regards, 

on behalf of

Assoc. Prof. Mehmet Demirci 

Academic Editor

PLOS ONE